# SSL-BN: Self-Supervised Learning Based on Structural Similarities in Brain Networks

## Abstract

Functional magnetic resonance imaging (fMRI) data provide critical information for the diagnosis of neurological disorders, as correlations among features of different regions of interest (ROIs) capture functional characteristics of the brain. Brain networks are an effective modeling paradigm for fMRI data, and recent works have explored GNN-based and Transformer-based approaches for brain network analysis. However, the dense and weighted edge structure of brain networks poses challenges for GNN-based methods, while Transformer-based methods typically require large amounts of labeled data. To address these issues, we propose a **S**elf-**S**upervised **L**earning framework for **B**rain **N**etworks (SSL-BN). Our approach pretrains a Brain Network Transformer by dispersing sample embeddings and refining them with a fixed, non-trainable matrix derived from a novel structural similarity measure, enabling contrastive representation learning without data augmentation. To our knowledge, SSL-BN is the first self-supervised framework specifically designed for brain networks. It employs a simple loss function, eliminates the need for augmentation, and significantly improves model performance on limited labeled data. Extensive experiments on the publicly available ABIDE dataset demonstrate that SSL-BN achieves state-of-the-art performance compared to prior methods.

## 1 Introduction

The brain network plays a crucial role in the diagnosis of neurological disease, with functional magnetic resonance imaging (fMRI) serving as a key technique. fMRI partitions the brain into regions of interest (ROIs) and records the blood oxygen level–dependent (BOLD) signals of each region in the form of time series. Since neurological disease can alter information transmission between different brain regions, pairwise correlations of BOLD signals across ROIs can provide valuable information for disease diagnosis. Such characteristics make brain network–based diagnosis particularly suitable for graph-related machine learning methods. Graph neural network (GNN)-based methods model ROIs as graph nodes and construct edges based on pairwise correlations. After this, Transformer-based methods introduce a self-attention mechanism to learn the relationships among different ROIs.

However, brain networks differ from typical graph structures in graph neural network applications, as they involve a larger number of nodes (ROIs) and edge connections. Moreover, supervised graph representation learning suffers from the excessive demand for labeled brain network data. These problems limit the performance of existing approaches in brain network-based diagnosis. To address this challenge, researchers have commonly adopted self-supervised pretraining to enhance the representation learning ability of graph models. These methods rely heavily on the design of graph augmentations, namely strategies that generate similar graph instances by perturbing edge connections in the input graph, in order to construct contrastive learning objectives. However, since brain networks are fully connected graphs and each edge carries a unique measurement (pairwise correlation), it becomes difficult to design effective augmentation strategies that produce meaningful graph instances.

In this work, we introduce the first **S**elf-**S**upervised **L**earning framework tailored for **B**rain **N**etwork–based diagnosis (SSL-BN), which avoids the need for graph augmentations and instead constructs learning objectives by exploring structural similarities among graph instances within the

dataset. We construct a *Dataset Graph* $G^{\mathbb{D}}$ by treating each brain network instance $g$ in the dataset as a graph node, where a *structural similarity* measure $\text{struct\_sim}(g_i, g_j)$ is designed to quantify the association between every brain network pair $(g_i, g_j)$ and serves as the feature of the corresponding edge $(i, j)$ in $G^{\mathbb{D}}$. Inspired by SGRL He et al. (2024), we then perform embedding dispersion and contrastive refinement on $G^{\mathbb{D}}$. Specifically, a dispersion loss function encourages normalized embeddings of $\{g_i\}$ to be dispersed on a unit hypersphere, then a non-trainable parameter matrix related to structural similarity is applied to linearly transform the dispersed embeddings, such that embeddings of more similar samples are pulled closer together, while embeddings of dissimilar samples are pushed apart. Through this process, the representation of each instance $g$ in the dataset is effectively pretrained, which enables the model to achieve competitive diagnostic accuracy even when fine-tuned with a relatively small number of labeled samples. Moreover, our pretraining framework entirely avoids the use of graph augmentations and achieves effective representation learning with only a single loss function. We primarily evaluate SSL-BN on the ABIDE dataset by comparing against supervised GNNs and self-supervised baselines. Additional experiments on the publicly available ADNI dataset, reported in the appendix, show our method's applicability and limitations under different conditions.

Our contribution is summarized as follows:

- We design a *Structural Similarity* measure $\text{struct\_sim}$ to construct the *Dataset Graph $G^{\mathbb{D}}$*. This enables a clear and effective evaluation of the similarity among brain network samples in the dataset, avoiding the generation of unreasonable augmentations for contrastive learning.

- We perform contrastive refinement of the embeddings by multiplying a non-trainable parameter matrix, thereby achieving effective pretraining without a complicated loss function.

- We conduct extensive experiments on the ABIDE dataset, supported by additional analysis on ADNI, to demonstrate the effectiveness and robustness of SSL-BN; additionally, ablation studies on the ABIDE dataset further validate the effectiveness of SSL-BN's design.

## 2 RELATED WORKS

### 2.1 GRAPH NEURAL NETWORKS AND TRANSFORMER-BASED METHODS

Graph Neural Networks (GNNs) are a class of machine learning methods designed for graph-structured data. They take node features and the adjacency matrix as input and learn node or graph representation through message passing between nodes. In real-world applications, graph structures can be constructed across different regions in various domains, which makes GNNs applicable to traffic forecasting Wang et al. (2020); Li & Zhu (2021); Ta et al. (2022), social networks Fan et al. (2019); Kumar et al. (2022); Guo et al. (2022), chemical molecules Wang et al. (2022); Gasteiger et al. (2021), and brain networks Li et al. (2021); Cui et al. (2022); Zhang et al. (2022); Wein et al. (2021) as studied in this work. Among commonly used GNN methods, Graph Convolutional Networks (GCNs) Kipf (2016) introduce normalization of the adjacency matrix and perform the summation of neighbor node features during message passing, while Graph Sample and Aggregate (GraphSAGE) Hamilton et al. (2017) samples a subset of neighbor nodes for feature learning and aggregation. With the advancement of research, Transformer-based methods have also been incorporated into graph learning, where the attention mechanism, widely used in natural language processing and computer vision Vaswani et al. (2017); Dosovitskiy et al. (2020); Liu et al. (2021), is leveraged to explore relationships between nodes. Graph Attention Networks (GATs) Veličković et al. (2017) employ trainable parameters to learn node similarities for message passing. Brain Network Transformer (BrainNetTF) Kan et al. (2022) adopts the Transformer architecture to model correlation matrices of brain networks. Due to its strong performance in brain network analysis, we adopt BrainNetTF as the encoder in our method.

### 2.2 GRAPH SELF-SUPERVISED LEARNING

Self-supervised learning is a paradigm that constructs training objectives without relying on sample labels, thereby enhancing the model's ability to learn from datasets where labels are scarce. Within

self-supervised learning, contrastive learning plays a particularly important role. It typically generates new augmentations by making slight modifications to existing samples and assumes that these augmented samples are similar to their originals in the representation space, while being dissimilar to negatives. The model then learns by optimizing a contrastive loss based on this assumption. Contrastive learning has been widely applied in computer vision, with representative methods including MoCo He et al. (2020), SimCLR Chen et al. (2020), and DINO Caron et al. (2021).

Self-supervised learning has also been applied in graph-related scenarios. For instance, Goodfellow et al. (2016); Zhu et al. (2020; 2021); Thakoor et al. (2021); He et al. (2024) explore graph connections and employ contrastive learning for node-level embedding in a self-supervised manner, where the model input is typically a single large graph. In contrast, for graph-level prediction tasks where datasets consist of multiple graphs, self-supervised pretraining is often followed by supervised fine-tuning with a small set of samples to capture the overall data distribution. InfoGraph Sun et al. (2019) achieves self-supervised learning by maximizing the mutual information between graph-level embeddings. GraphLoG Xu et al. (2021) utilizes an EM algorithm Dempster et al. (1977) to incorporate both local and global structures. D-SLA Kim et al. (2022) introduces a loss function opposite to contrastive loss and leverages a discriminator to distinguish between different graph augmentations. GraphSSL Zeng & Xie (2021) constructs augmentations by removing and adding nodes or edges. However, these existing methods are designed for generic graph inputs and are not well-suited for brain networks. Therefore, in this work, we propose SSL-BN, a self-supervised learning framework specifically tailored to the characteristics of brain networks.

## 3 METHODOLOGY

In this study, we propose a self-supervised learning approach that enhances brain network representations, thereby improving the performance of fMRI-based disease diagnosis.

### 3.1 PROBLEM SETTING

When working with fMRI data, preprocessing is an essential step. We denote the preprocessed dataset as $\mathbb{D} = \{g_i | i = 1, ..., N\}$, where each $g_i$ corresponds to an fMRI scan $f_i$. We define the brain network representation as follows:

$$e_i = \text{Encoder}(g_i). \tag{1}$$

Typically, an fMRI scan $f$ is partitioned into $n$ Regions of Interest (ROIs). The BOLD signal of each region $r$ is extracted and denoted by $t_r(r = 1, ..., n)$. We use these BOLD signals to compute the correlation profile of $f$, obtaining a correlation matrix $\mathbf{C} \in \mathbb{R}^{n \times n}$:

$$\mathbf{C}[r, s] = \text{corr}(t_r, t_s), \tag{2}$$

where corr represents the Pearson correlation function. Since $\mathbf{C}$ reflects the connectivity characteristics of the brain network, we define $g_i$ as $\mathbf{C}^i$ for representation learning and subsequently predict whether an fMRI scan is in a diseased or normal state.

Self-supervised learning constructs training objectives without relying on ground-truth labels, enabling the encoder to learn representations from the data's inherent structure. In this paper, we design a pretraining framework based solely on $\{e_i\}$ and $\{g_i\}$, enabling the model to achieve satisfactory performance even when trained on a limited number of labeled samples.

### 3.2 METHOD OVERVIEW

The overview of our SSL-BN framework is illustrated in Fig. 1. Given a dataset $\mathbb{D}$ of length $N$, we design a structural similarity computation method to quantify the degree of similarity between each pair of data samples. Based on this similarity measure, we construct a dataset graph $G^{\mathbb{D}}$ with specific edge features. We then apply the loss function so that, after normalization, all data embeddings are constrained to lie on a hypersphere while maintaining a considerable degree of dispersion from each other. Finally, by employing a simple matrix multiplication operation, we bring the embeddings of similar data samples closer together, while simultaneously pushing apart the embeddings of dissimilar data samples.

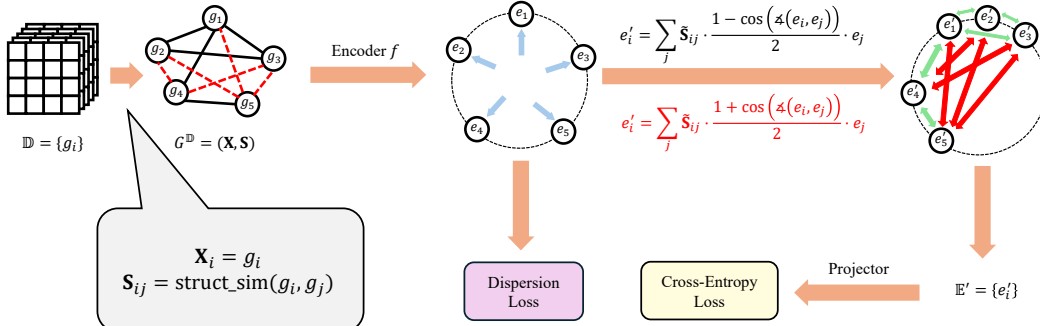

Figure 1: Overview of SSL-BN. We first construct a dataset graph using all samples in the dataset, where the edge features are defined by the structural similarity between each pair of samples. Each sample's embedding is then normalized and dispersed onto a hypersphere by applying a dispersion loss. Subsequently, we perform a non-trainable refinement based on centralized similarity and the angular relationships between embeddings, encouraging similar samples to move closer together in the representation space while pushing dissimilar ones farther apart. The refined embeddings are then used to perform the final classification task.

For encoding, we adopt the Brain Network Transformer Kan et al. (2022), which has demonstrated state-of-the-art performance in fMRI-based diagnosis. This model takes the correlation matrix $\mathbf{C}$ as input and incorporates a self-attention mechanism, making it the most effective brain network encoder to date.

### 3.3 STRUCTURAL SIMILARITY AND DATASET GRAPH

To investigate the inter-sample associations within the dataset, we design a measure for the correlation matrix instances to examine the structural similarity of brain networks. We categorize the relationships between ROI $i$ and ROI $j$ based on $\mathbf{C}[i,j] = c_{ij}$ into three groups:

- Negative correlated (NC): $c_{ij} < -\theta$,
- Uncorrelated (UC): $-\theta \le c_{ij} \le \theta$,
- Positive correlated (PC): $c_{ij} > \theta$.

Given two matrices $\mathbf{C}^m, \mathbf{C}^n$, if $c_{ij}^m$ and $c_{ij}^n$ fall into the same category, their similarity should increase; conversely, when they belong to different categories, their similarity decreases. However, the influence of different cases on similarity should not be regarded as equal. For example, when $c_{ij}^m$ and $c_{ij}^n$ belong to NC and PC, respectively, the reduction in similarity should be greater than the case where one of them is UC. Similarly, when both are NC or both are PC, the contribution to similarity should be greater than the case where both are UC. Therefore, we define the computation of structural similarity as follows:

$$\text{struct\_sim}(\mathbf{C}^m, \mathbf{C}^n) = \sum_{i,j} f_{corr}(c_{ij}^m, c_{ij}^n) \tag{3}$$

$$f_{corr}(x,y) = \begin{cases} \alpha_1, & x, y \text{ are both NC or PC}, \\ \alpha_2, & x, y \text{ are both UC}, \\ \alpha_3, & x, y \text{ are NC and PC/PC and NC respectively}, \\ \alpha_4, & \text{otherwise}, \end{cases} \tag{4}$$

where $\alpha_1 > \alpha_2 > 0 > \alpha_4 > \alpha_3$. Through this computation, we construct a similarity matrix $\mathbf{S} \in \mathbb{R}^{N \times N}$ for the dataset $\mathbb{D}$ as follows:

$$\mathbf{S}[m,n] = \text{struct\_sim}(\mathbf{C}^m, \mathbf{C}^n). \tag{5}$$

If we treat all samples in the dataset as nodes to construct a graph, then $S$ serves as the edge features. We denote such a graph as the *Dataset Graph* $G^{\mathbb{D}} = (\mathbb{D}, \mathbf{S})$.

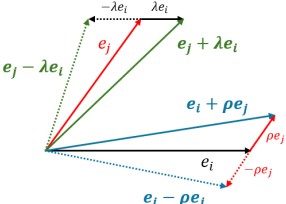

Figure 2: Visualization of embedding addition and subtraction. Addition reduces the angular distance between embeddings, whereas subtraction increases it.

### 3.4 EMBEDDING DISPERSION AND CONTRASTIVE REFINEMENT

In SGRL, to obtain node-level embeddings of a graph $G'$ via contrastive learning, the authors first normalize all embeddings and distribute them on a unit hypersphere, denoted as $\mathbf{E} = [e_1, e_2, ..., e_{|G'|}]^T \in \mathbb{R}^{|G'| \times C}$. Then, through matrix multiplication with the adjacency matrix $\mathbf{A} \in \mathbb{R}^{|G'| \times |G'|}$, the new embeddings (denoted as $\mathbf{E}' \in \mathbb{R}^{|G'| \times C}$) of adjacent nodes become closer in the representation space. Specifically, the procedure is as follows:

$$\mathbf{E}' = \mathbf{A}\mathbf{E} + \mathbf{E}. \tag{6}$$

If we set $G'$ as the $G^{\mathbb{D}}$ constructed in the previous section, we can, by a similar procedure, gather the embeddings of nodes (i.e., data samples in $\mathbb{D}$) with higher structural similarities together, and make dissimilar data samples further apart. In this way, the dispersed embeddings can be contrastively refined solely through matrix multiplication, ensuring the new embedding $\mathbf{E}'$ is more suitable for the classification task.

#### 3.4.1 EMBEDDING DISPERSION BY CENTER SUPERPOSITION

First, to sufficiently disperse $\{e_i\}$, we apply $l_2$ normalization and then compute the sum of the distances from each sample's embedding to their centroid $c$, and design a dispersion loss based on this quantity so that $c$ coincides with the center of the unit hypersphere. Specifically, it is formulated as follows:

$$L_{\mathrm{disp}} = -\frac{1}{N} \sum_{i=1}^{N} ||\tilde{e}_i - c||_2^2, \quad c = \frac{1}{N} \sum_{i=1}^{N} \tilde{e}_i, \tag{7}$$

where $\tilde{e}_i = e_i/(||e_i||_2)$. Clearly, this loss function converges to $-1$, at which point all $\tilde{e}_i$ are mutually dispersed on the unit hypersphere. Due to the use of normalization, the $l_2$ norm of each $\tilde{e}i$ always remains 1, which also ensures rapid convergence of $L_{\mathrm{disp}}$. We denote the dispersed embeddings as $\mathbf{E} = [e_1, e_2, ..., e_N]^T \in \mathbb{R}^{N \times C}$.

#### 3.4.2 CONTRASTIVE REFINEMENT BY LINEAR TRANSFORMATION

In general, given two embeddings $e_i$ and $e_j$, stretching one embedding toward the direction of the other reduces the angle between them. Conversely, stretching it in the opposite direction increases the angle. Fig. 2 illustrates this process. Therefore, if we apply a refinement to each $e_i$ such that it is slightly stretched toward all embeddings similar to it, and slightly stretched in the opposite direction of embeddings dissimilar to it, we can construct, in a contrastive manner, a new set of embeddings $\{e_i'\}$. This procedure assembles samples with higher structural similarity while separating clusters with low structural similarity from each other. In this way, the subsequent projector can perform the classification task more effectively.

According to Fig. 2, we observe that the refinement of embeddings can be formulated in matrix multiplication as follows:

$$\mathbf{E}' = \mathbf{\Lambda}\mathbf{E}, \tag{8}$$

where diagonal elements of $\mathbf{\Lambda}$ are set to 1. This computation is similar to Eq. 6; however, in Eq. 6, $\mathbf{A}$ is binary, which implies that there are no negative values to control the opposite-direction stretching of embeddings, and no distinction is made regarding the degree of refinement (stretching) for

different embeddings. To address these problems, in our method, the similarity matrix $\mathbf{S}$ derived in Sec. 3.3 will be applied to calculate the values in $\mathbf{\Lambda}$.

Intuitively, the more similar two samples $g_i$ and $g_j$ are, the larger $\mathbf{S}[i, j]$ is, and correspondingly, $\mathbf{\Lambda}[i, j]$ should also be larger and positive. Conversely, the less similar the two samples are, the smaller $\mathbf{S}[i, j]$ is, and $\mathbf{\Lambda}[i, j]$ should become smaller and negative. Moreover, when $|\mathbf{\Lambda}[i, j]| = |\mathbf{\Lambda}[j, i]| = 1$, the angle between $e'_i$ and $e'_j$ will become $180°$ or $0°$ (only considering the stretching of the two embeddings). Therefore, it is necessary to centralize $\mathbf{S}$ and divide it by its range. Specifically:

$$\tilde{\mathbf{S}} = \frac{\mathbf{S} - \bar{\mathbf{S}}}{\max(\mathbf{S}) - \min(\mathbf{S})}. \tag{9}$$

After considering the influence of $\mathbf{S}$ on $\mathbf{\Lambda}$, we further analyze the problem from the perspective of $\mathbf{E}$. When the angle between $e_i$ and $e_j$ is small, if $\tilde{\mathbf{S}}[i, j] > 0$, then they do not need to be stretched too much, and thus $|\mathbf{\Lambda}[i, j]|$ should approach 0. Similarly, when the angle between $e_i$ and $e_j$ is large, if $\mathbf{S}[i, j] < 0$, then they do not need to be pushed much farther apart, and thus $|\mathbf{\Lambda}[i, j]|$ should also approach 0. Mathematically, when the angle is small, $||e_i + \mathbf{\Lambda}[i, j]e_j||_2 \approx ||e_i|| + |\mathbf{\Lambda}[i, j]| \cdot ||e_j||$; when the angle is close to $180°$, $||e_i - \mathbf{\Lambda}[i, j]e_j||_2 \approx ||e_i|| + |\mathbf{\Lambda}[i, j]| \cdot ||e_j||$. Therefore, the benefit of controlling the value of $\mathbf{\Lambda}[i, j]$ in these cases is that it prevents an exponential growth of $||e'_i||_2$, which is detrimental to the projector in accomplishing the final classification task.

To address this issue, we design a parameter matrix $\mathbf{\Omega}$ to constrain the elements in $\tilde{\mathbf{S}}$. Specifically, when $\tilde{\mathbf{S}}[i, j] > 0$, $\mathbf{\Omega}[i, j]$ decreases as $\angle(e_i, e_j)$ becomes smaller; when $\tilde{\mathbf{S}}[i, j] < 0$, $\mathbf{\Omega}[i, j]$ decreases as $\angle(e_i, e_j)$ becomes larger. Furthermore, the values in $\mathbf{\Omega}$ should remain within the range $[0, 1]$. Based on the above requirements, we make the following design:

$$\mathbf{\Omega}[i, j] = \begin{cases} 0.5(1 - \cos\_\mathrm{sim}(e_i, e_j)), & \tilde{\mathbf{S}}[i, j] > 0, \\ 0.5(1 + \cos\_\mathrm{sim}(e_i, e_j)), & \tilde{\mathbf{S}}[i, j] < 0, \\ 0, & \text{otherwise}, \end{cases} \tag{10}$$

where $\cos\_\mathrm{sim}$ is the cosine similarity. In combination with Eq. 8, we derive the final computation formula for the contrastive refinement:

$$\mathbf{\Lambda} = \tilde{\mathbf{S}} \odot \mathbf{\Omega} \quad \Rightarrow \quad \mathbf{E}' = (\tilde{\mathbf{S}} \odot \mathbf{\Omega})\mathbf{E}, \tag{11}$$

where $\odot$ is the element-wise multiplication.

In our method, only the dispersion loss is employed during pretraining, while contrastive refinement does not require training and is therefore typically applied before the dispersed embeddings are passed into the projector during finetuning. Considering the limitation of computational resources, supervised learning typically adopts a mini-batch strategy. Therefore, we treat each batch as a set and construct a batch graph in a manner similar to the dataset graph. Because batches are randomly sampled in each epoch during training, the embeddings undergo repeated contrastive refinement in different batches. Consequently, the overall effect gradually approximates that of the pretraining stage, where the dataset graph is employed.

## 4 EXPERIMENTS

In this section, we apply our SSL-BN method to pretrain the encoder on the ABIDE and ADNI datasets. We then evaluate its performance by training with a small proportion of samples and testing on a larger set, reporting the corresponding metrics. In parallel, we adopt a similar experimental setup to evaluate other GNN-based and self-supervised methods on the same datasets. Finally, we design and conduct a series of ablation studies to demonstrate the soundness and effectiveness of our proposed design.

### 4.1 DATASET

The *Autism Brain Imaging Data Exchange* (ABIDE) Craddock et al. (2013) dataset provides resting-state functional magnetic resonance imaging (rs-fMRI) data from 17 international sites, followed by functional parcellation using the Craddock 200 Craddock et al. (2012) atlas. The dataset contains a

total of 1009 samples, among which 516 are labeled as Autism Spectrum Disorder (ASD), and the remaining are control samples. After parcellation, each sample consists of 200 Regions of Interest (ROIs), and each ROI is associated with a BOLD signal time series. Thus, the resulting correlation matrix has a size of $200 \times 200$. Due to its public availability and standardized preprocessing, this dataset is widely used in studies related to brain disease diagnosis.

The *Alzheimer's Disease Neuroimaging Initiative* (ADNI) dataset is a large-scale, longitudinal project designed to investigate biomarkers for the early detection and progression of Alzheimer's disease (AD). It provides multimodal neuroimaging data, including structural magnetic resonance imaging (sMRI), functional MRI (fMRI), and positron emission tomography (PET), along with clinical and cognitive assessments. In our work, raw DICOM data were converted to the Brain Imaging Data Structure (BIDS) standard Gorgolewski et al. (2016) using dcm2bids v2.1.4 Boré et al. (2023). Preprocessing of the fMRI data was performed with fMRIPrep v24.1.1 Esteban et al. (2019), which included slice-timing correction, motion correction, susceptibility distortion correction, and spatial normalization to standard space. Functional connectivity correlation matrices for each subject were generated with XCP-D v0.10.5 Ciric et al. (2023) using 131 regions of interest derived from a combined cortical 17-network parcellation Yeo et al. (2011) and cerebellar 17-network parcellation Buckner et al. (2011). This enables the input to be represented in the form of a $131 \times 131$ matrix. We extract 102 Alzheimer's Disease (AD) and 102 (CN) samples for our experiments.

## 4.2 IMPLEMENTATION DETAILS AND EVALUATION METRICS

In this study, all self-supervised training procedures are conducted using the entire dataset without leveraging any label information. For experiments involving supervised learning, the dataset is randomly split into training, validation, and testing sets with a ratio of $2 : 1 : 7$. Each evaluation metric is displayed in the form of mean and standard deviation over five repeated runs of training, validation, and testing, where all runs are performed under the same dataset partitioning.

In the computation of structural similarity in Eq. 4, we categorize the correlation values into three classes—NC, UC, and PC—based on the parameter $\theta$. In this work, we set $\theta = 0.3$, which was found to yield the best results after extensive experimentation. Subsequently, we perform classification and summation according to Eq. 3. To ensure computational efficiency on the GPU, we implement the above procedure using matrix multiplication. Specifically, we define the following three matrices:

$$\mathbf{P_1}[i,j] = \begin{cases} 1, & \mathbf{\Gamma}^{\mathbb{D}}[i,j] \text{ is PC}, \\ 0, & \mathbf{\Gamma}^{\mathbb{D}}[i,j] \text{ is UC}, \\ -1, & \mathbf{\Gamma}^{\mathbb{D}}[i,j] \text{ is NC}, \end{cases} \tag{12}$$

$$\mathbf{P_2}[i,j] = \begin{cases} 1, & \mathbf{\Gamma}^{\mathbb{D}}[i,j] \text{ is PC}, \\ -0.5, & \mathbf{\Gamma}^{\mathbb{D}}[i,j] \text{ is UC}, \\ 0, & \mathbf{\Gamma}^{\mathbb{D}}[i,j] \text{ is NC}, \end{cases} \quad \mathbf{P_3}[i,j] = \begin{cases} 0, & \mathbf{\Gamma}^{\mathbb{D}}[i,j] \text{ is PC}, \\ 0.5, & \mathbf{\Gamma}^{\mathbb{D}}[i,j] \text{ is UC}, \\ -1, & \mathbf{\Gamma}^{\mathbb{D}}[i,j] \text{ is NC}, \end{cases} \tag{13}$$

, where

$$\mathbf{\Gamma}^{\mathbb{D}}[m] = \text{flatten}(\mathbf{C}^m) \in \mathbb{R}^{N^2}. \tag{14}$$

In this way, the similarity matrix can be calculated as follows:

$$\mathbf{S} = \mathbf{P_1}\mathbf{P_1}^T + \mathbf{P_2}\mathbf{P_2}^T + \mathbf{P_3}\mathbf{P_3}^T. \tag{15}$$

The coefficient 0.5 in Eq. 13 indicates that, when computing element-wise similarity, the negative impact on the overall similarity should be smaller if one of the two elements belongs to the UC class; furthermore, when both elements are UC, the positive contribution to the overall similarity should be even smaller. Therefore, computing $\mathbf{S}$ using Eq. 15 is equivalent to using Eq. 3 and Eq. 4, where the coefficients in Eq. 4 are given as $\alpha_1 = 2, \alpha_2 = 0.5, \alpha_3 = -1, \alpha_4 = -0.5$.

In the embedding dispersion stage, since we only have a single loss term $L_{\text{disp}}$ and this process does not involve learning information from the matrix input itself, it is unnecessary to train for a large number of epochs. We adopt the Adam optimizer Kingma (2014) with the learning rate = 0.0001 and the number of epochs per repeat = 50. However, these hyperparameter choices are not critical, as $L_{\text{disp}}$ consistently converges to a value very close to $-1$ during training regardless of the settings. This property highlights the robustness and stability of our design.

| Method | Accuracy | AUROC | Sensitivity | Specificity |
|---|---|---|---|---|
| GCN | $63.25 \pm 1.19$ | $68.01 \pm 1.42$ | $67.25 \pm 3.64$ | $59.09 \pm 3.34$ |
| GAT | $59.41 \pm 1.26$ | $63.34 \pm 0.49$ | $59.29 \pm 1.36$ | $59.29 \pm 1.36$ |
| GraphSAGE | $59.69 \pm 3.22$ | $65.87 \pm 1.54$ | $77.09 \pm 6.21$ | $41.72 \pm 12.28$ |
| InfoGraph | $51.46 \pm 1.14$ | $50.26 \pm 2.23$ | $51.46 \pm 1.20$ | $51.65 \pm 4.99$ |
| SGRL | $62.80 \pm 4.53$ | $69.39 \pm 6.72$ | $70.91 \pm 12.72$ | $54.85 \pm 9.59$ |
| SimCLR[1] | $65.20 \pm 5.98$ | $71.18 \pm 4.92$ | $72.32 \pm 18.07$ | $57.60 \pm 20.56$ |
| SimCLR[2] | $63.00 \pm 3.46$ | $70.57 \pm 5.12$ | $62.79 \pm 18.34$ | $62.99 \pm 21.41$ |
| MoCo[1] | $61.40 \pm 7.00$ | $70.66 \pm 5.74$ | $\mathbf{84.62 \pm 11.42}$ | $35.75 \pm 19.10$ |
| MoCo[2] | $64.60 \pm 2.24$ | $70.78 \pm 3.89$ | $59.02 \pm 14.35$ | $\mathbf{70.62 \pm 15.22}$ |
| **SSL-BN** | $\mathbf{67.80 \pm 6.18}$ | $\mathbf{73.73 \pm 6.02}$ | $67.30 \pm 6.81$ | $68.58 \pm 7.80$ |
| BrainNetTF | $65.40 \pm 2.87$ | $72.51 \pm 3.18$ | $70.46 \pm 14.51$ | $59.10 \pm 13.12$ |

Table 1: Comparison experimental results of different methods on ABIDE dataset.

For fine-tuning after model pretraining, we adopt the default settings of BrainNetTF, using the Adam optimizer Kingma (2014) with a batch size of 64 and 200 training epochs. Since the training set is small, we employ a smaller learning rate of $10^{-5}$, along with a weight decay of $10^{-5}$.

Regarding the evaluation of experimental results, we report the test accuracy, area under the receiver operating characteristic curve (AUROC), sensitivity, and specificity. Since AUROC considers different thresholds in binary classification tasks, we adopt it as the primary reference metric and use the training parameters corresponding to the highest validation AUROC during testing.

### 4.3 Comparison Experiments

In this section, we present the performance metrics of our SSL-BN compared with other baseline approaches on the ABIDE and ADNI datasets. Since there have been no existing self-supervised learning methods specifically designed for brain networks, we applied the following three categories of methods for comparison with our proposed approach: supervised GNN methods, Graph self-supervised methods, and other contrastive learning methods. Table 1 report the evaluation metrics of all methods, and the last section of the tables presents the results of our SSL-BN method as well as the BrainNetTF method without self-supervised pretraining, demonstrating the effectiveness of SSL-BN pretraining. Experimental results on the ADNI dataset are presented in Appendix A.1.

### 4.3.1 Supervised GNN methods

Since the connectivity of brain networks can be utilized to construct graph edges, we experimented with three commonly used GNN methods, namely GCN, GAT, and GraphSAGE. These models were trained, validated, and tested on the dataset split in a 2:1:7 ratio. For each sample, the adjacency matrix $\mathbf{A}$ was derived from the correlation matrix $\mathbf{C}$, specifically defined as $\mathbf{A} = \mathbf{1}_{\{\mathbf{C} > \text{threshold}\}}$. The model input was given as $(\mathbf{C}, \mathbf{A})$. The evaluation results are reported in the first part of Table 1, which demonstrate that GNN methods exhibit limited capability in learning from brain networks, particularly when training samples are scarce. This limitation arises from the fact that brain network connectivity is not inherently binary, and simple thresholding inevitably leads to a partial loss of structural information in the input graph. Appendix A.2 presents the experimental results of GNN methods under different threshold values, while the results reported in Table 1 correspond to the best-performing threshold.

### 4.3.2 Graph Self-supervised Methods

In the second section of the tables, we present the test results of recent graph self-supervised learning methods on the two datasets. The results indicate that SSL-BN outperforms these other methods. This is because most of those methods are designed for graph inputs, not brain networks. In contrast, our method performs contrastive refinement with a structural similarity computation specifically tailored for brain networks, which makes our method more suitable.

### 4.3.3 BASELINE CONTRASTIVE LEARNING METHODS

Since the input of a brain network (the correlation matrix) is a 2D array, similar to an image, we also evaluated SimCLR Chen et al. (2020) and MoCo He et al. (2020), which are widely used contrastive learning methods in computer vision. In addition, we employed a common augmentation strategy from computer vision, namely randomly masking elements of the correlation matrix to zero. We explored two approaches: (1) applying random masking to the input BOLD signals and then computing the correlation matrix as the augmentation result, and (2) directly applying random masking to the correlation matrix itself as the augmentation. The results of SimCLR and MoCo are presented in the third section of the tables. These results highlight the difficulty of generating meaningful augmentations for brain networks, as the information in the correlation matrix is highly dense, and random masking in either approach cannot reliably produce truly similar samples. Appendix A.3 presents the experimental results of SimCLR and MoCo under different masking ratios, while the results reported in Table 1 correspond to the best-performing masking ratio.

## 4.4 ABLATION STUDIES

To validate the effectiveness of our method, we replaced several key components of SSL-BN and ran experiments on the ABIDE dataset. The results are reported in Table 2. First, we remove the embedding dispersion step and apply contrastive refinement directly after encoder initialization. We observed a performance drop, which can be attributed to the difficulty of separating embeddings of dissimilar samples solely through the linear transformation in contrastive refinement without first dispersing them. Next, we replaced our structural similarity computation with the commonly used cosine similarity. As shown in the table, this substitution led to inferior results, further demonstrating that our design is better suited for brain networks. We then modified Eq. 11 by replacing the parameter matrix $\tilde{\mathbf{S}} \odot \mathbf{\Omega}$ with either $\tilde{\mathbf{S}}$ or $\mathbf{\Omega}$ alone. However, the experimental results indicate that the best performance is achieved only when both matrices are used together. Finally, we removed the entire contrastive refinement component and used only the dispersed embeddings for model finetuning. The results confirm that our contrastive refinement plays a crucial role in enhancing model performance.

| Variant | Accuracy | AUROC | Sensitivity | Specificity |
|---|---|---|---|---|
| SSL-BN w/o dispersion | $62.00 \pm 2.10$ | $66.32 \pm 3.04$ | $61.48 \pm 13.61$ | $62.49 \pm 15.90$ |
| SSL-BN w/ cos similarity | $64.00 \pm 7.64$ | $67.20 \pm 5.65$ | $\mathbf{74.14 \pm 17.22}$ | $53.44 \pm 26.15$ |
| SSL-BN w/ only $\tilde{\mathbf{S}}$ | $62.60 \pm 5.54$ | $69.95 \pm 3.44$ | $55.35 \pm 8.94$ | $\mathbf{70.45 \pm 4.69}$ |
| SSL-BN w/ only $\mathbf{\Omega}$ | $58.80 \pm 3.87$ | $64.54 \pm 4.29$ | $59.14 \pm 11.97$ | $59.78 \pm 9.67$ |
| SSL-BN w/o refinement | $64.40 \pm 6.41$ | $67.08 \pm 6.47$ | $70.95 \pm 9.94$ | $57.14 \pm 13.58$ |
| **SSL-BN** | $\mathbf{67.80 \pm 6.18}$ | $\mathbf{73.73 \pm 6.02}$ | $67.30 \pm 6.81$ | $68.58 \pm 7.80$ |

Table 2: Ablation studies of SSL-BN.

## 5 CONCLUSION

In this paper, we present SSL-BN, a self-supervised learning framework specifically developed for brain networks. Our method adopts BrainNetTF, a highly effective encoder for brain network analysis, and takes the correlation matrix between brain ROIs as input. First, we normalize the embeddings of each sample in the dataset and disperse them onto a unit hypersphere. Next, we design a structural similarity calculation tailored to brain networks to capture inter-sample relationships, and construct a parameter matrix for linear transformation based on both this similarity and the pairwise angular relationships between embeddings. The dispersed embeddings are then refined through simple matrix multiplication. In this way, we pretrain the encoder with a straightforward process that removes the need for augmentation. During finetuning, we repeat the same procedure within each minibatch. Our experimental results demonstrate that SSL-BN achieves state-of-the-art performance compared to various baselines, and our ablation studies further validate the effectiveness of the proposed design.

## 6  REPRODUCIBILITY STATEMENT

The code will be released publicly upon acceptance. We adopt a single NVIDIA RTX A6000 GPU for all the experiments.

## 7  ETHICS STATEMENT

This work uses only publicly available datasets. This study did not involve the collection of new data from human participants.

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

# A  APPENDIX

## A.1  COMPARISON EXPERIMENTAL RESULTS OF ADNI DATASET

Table 3: Comparison Experimental Results of ADNI Dataset

| Method | Accuracy | AUROC | Sensitivity | Specificity |
|---|---|---|---|---|
| GCN | $65.42 \pm 1.72$ | $70.92 \pm 1.72$ | $53.68 \pm 5.93$ | $73.10 \pm 2.00$ |
| GAT | $\mathbf{69.86 \pm 2.66}$ | $\mathbf{73.86 \pm 3.90}$ | $50.87 \pm 3.53$ | $\mathbf{81.17 \pm 3.42}$ |
| InfoGraph | $56.10 \pm 1.68$ | $55.25 \pm 3.98$ | $56.10 \pm 1.68$ | $52.68 \pm 4.84$ |
| **SSL-BN** | $60.00 \pm 3.16$ | $69.55 \pm 2.99$ | $\mathbf{77.17 \pm 18.32}$ | $50.86 \pm 22.38$ |
| BrainNetTF | $51.00 \pm 9.70$ | $62.93 \pm 7.37$ | $49.20 \pm 34.55$ | $60.95 \pm 26.94$ |

On the ADNI dataset, we evaluated several methods, including supervised graph-based approaches such as GCN and GAT, as well as the graph self-supervised learning method InfoGraph. As shown in the table, GCN and GAT outperform Transformer-based methods, primarily because the graphs in the ADNI dataset are relatively small and the overall number of samples is limited, which prevents Transformer-based models from fully demonstrating their advantages. Nevertheless, our SSL-BN still provides a clear performance boost to BrainNetTF, further validating the effectiveness of our approach.

## A.2  GCN, GAT AND GRAPHSAGE WITH DIFFERENT THRESHOLD ON ABIDE DATSAET

| Method | Accuracy | AUROC | Sensitivity | Specificity |
|---|---|---|---|---|
| GCN (0.9) | $62.74 \pm 1.96$ | $67.84 \pm 1.53$ | $67.17 \pm 2.57$ | $58.15 \pm 4.38$ |
| GCN (0.7) | $63.25 \pm 1.19$ | $68.01 \pm 1.42$ | $67.25 \pm 3.64$ | $59.09 \pm 3.34$ |
| GCN (0.5) | $62.43 \pm 1.72$ | $67.45 \pm 2.03$ | $63.66 \pm 3.53$ | $61.06 \pm 4.92$ |
| GCN (0.3) | $60.88 \pm 1.35$ | $65.23 \pm 1.75$ | $65.00 \pm 2.25$ | $56.60 \pm 3.46$ |
| GAT (0.9) | $58.33 \pm 1.40$ | $61.61 \pm 0.57$ | $61.61 \pm 0.57$ | $49.67 \pm 15.66$ |
| GAT (0.7) | $58.30 \pm 1.46$ | $61.84 \pm 1.56$ | $62.73 \pm 12.97$ | $53.81 \pm 13.79$ |
| GAT (0.5) | $58.59 \pm 0.23$ | $61.97 \pm 0.82$ | $55.49 \pm 10.89$ | $61.76 \pm 11.41$ |
| GAT (0.3) | $59.41 \pm 1.26$ | $63.34 \pm 0.49$ | $59.29 \pm 1.36$ | $59.29 \pm 1.36$ |
| GraphSAGE (0.9) | $58.19 \pm 2.89$ | $65.96 \pm 0.75$ | $75.63 \pm 16.78$ | $40.34 \pm 21.89$ |
| GraphSAGE (0.7) | $59.66 \pm 1.51$ | $65.41 \pm 1.27$ | $72.02 \pm 5.39$ | $46.89 \pm 7.88$ |
| GraphSAGE (0.5) | $59.69 \pm 3.22$ | $65.87 \pm 1.54$ | $77.09 \pm 6.21$ | $41.72 \pm 12.28$ |
| GraphSAGE (0.3) | $57.37 \pm 1.78$ | $64.59 \pm 0.87$ | $79.42 \pm 6.40$ | $34.44 \pm 9.98$ |

Table 4: GCN, GAT and GraphSAGE with different threshold on ABIDE datsaet. Thresholds are in the parentheses.

## A.3  MOCO AND SIMCLR WITH DIFFERENT MASKING RATIO ON ABIDE DATSAET

| Method | Accuracy | AUROC | Sensitivity | Specificity |
|--------|----------|-------|-------------|-------------|
| MoCo, 5% - 10% corr | $62.80 \pm 6.05$ | $70.16 \pm 4.37$ | $76.13 \pm 14.87$ | $49.40 \pm 5.63$ |
| MoCo, 5% - 10% TS | $62.60 \pm 2.06$ | $69.22 \pm 5.52$ | $84.04 \pm 11.34$ | $37.24 \pm 17.22$ |
| MoCo, 10% - 20% corr | $64.60 \pm 2.24$ | $70.78 \pm 3.89$ | $59.02 \pm 14.35$ | $70.62 \pm 15.22$ |
| MoCo, 10% - 20% TS | $61.40 \pm 7.00$ | $70.66 \pm 5.74$ | $84.62 \pm 11.42$ | $35.75 \pm 19.10$ |
| MoCo, 20% - 40% corr | $61.20 \pm 3.19$ | $67.85 \pm 3.84$ | $76.64 \pm 11.56$ | $48.30 \pm 12.18$ |
| MoCo, 20% - 40% TS | $62.20 \pm 2.14$ | $69.28 \pm 4.52$ | $72.48 \pm 11.69$ | $49.09 \pm 21.09$ |
| SimCLR, 5% - 10% corr | $63.20 \pm 4.66$ | $68.29 \pm 4.41$ | $73.97 \pm 13.28$ | $51.23 \pm 14.29$ |
| SimCLR, 5% - 10% TS | $65.20 \pm 5.98$ | $71.18 \pm 4.92$ | $72.32 \pm 18.07$ | $57.60 \pm 20.56$ |
| SimCLR, 10% - 20% corr | $63.60 \pm 3.93$ | $70.24 \pm 2.03$ | $69.34 \pm 16.80$ | $55.64 \pm 18.30$ |
| SimCLR, 10% - 20% TS | $64.40 \pm 4.50$ | $70.35 \pm 3.58$ | $66.58 \pm 13.42$ | $63.25 \pm 15.08$ |
| SimCLR, 20% - 40% corr | $63.00 \pm 3.46$ | $70.57 \pm 5.12$ | $62.79 \pm 18.34$ | $62.99 \pm 21.41$ |
| SimCLR, 20% - 40% TS | $65.00 \pm 5.22$ | $68.93 \pm 6.57$ | $81.67 \pm 11.55$ | $44.29 \pm 8.00$ |

Table 5: MoCo and SimCLR with different masking ratio on ABIDE dataset. "coor" means correlation matrix, and "TS" means BOLD time series.

## A.4 ROC CURVE OF SSL-BN AND BRAINNETTF

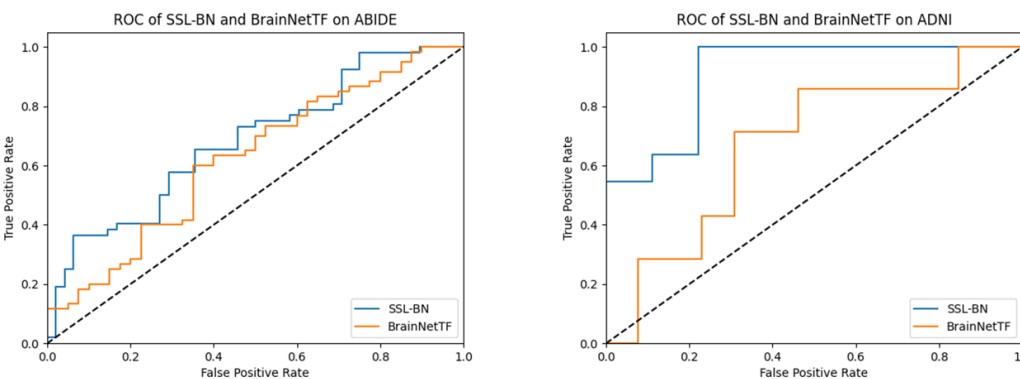

Figure 3: ROC curve of SSL-BN and BrainNetTF on ABIDE and ADNI dataset.

## A.5 LLM DISCLOSURE

Language models were used to assist with grammar and wording, and all content was carefully reviewed and validated by the authors.

