# OpenReview forum: "SSL-BN: Self-Supervised Learning Based on Structural Similarities in Brain Networks"
_ICLR.cc/2026/Conference — ICLR 2026 Conference Withdrawn Submission_

### Official Review · Reviewer_mcHs · 2025-10-21

**Soundness:** 2
**Presentation:** 2
**Contribution:** 2
**Rating:** 4
**Confidence:** 3

**Summary:**

This paper addresses the challenge that existing self-supervised methods are difficult to directly apply to brain functional network classification tasks. The authors propose a self-supervised pretraining framework specifically designed for brain networks, termed SSL-BN. The method disperses sample embeddings and refines them using a fixed, non-trainable matrix constructed from a novel structural similarity measure. This enables pretraining of a Transformer-based brain network feature extractor via contrastive representation learning without requiring data augmentation. The approach effectively avoids the potential risks of generating inappropriate augmented samples in contrastive learning. A series of comparative and ablation experiments are conducted to verify the effectiveness of the proposed framework.

**Strengths:**

The paper designs a contrastive learning method that does not rely on data augmentation, specifically tailored for brain network classification. By replacing traditional augmentation with a combination of structural similarity metric and non-trainable parameter matrix, the method mitigates the difficulty of generating meaningful augmented samples for brain networks. This breaks through the conventional paradigm of self-supervised learning that heavily depends on data augmentation.

**Weaknesses:**

1.The authors claim this is the first self-supervised framework specifically designed for brain networks. However, several prior studies, such as BrainMass: Advancing Brain Network Analysis for Diagnosis with Large-scale Self-Supervised Learning and Graph Self-Supervised Learning with Application to Brain Network Analysis, have already proposed self-supervised methods for brain networks. The manuscript does not sufficiently compare or clearly differentiate itself from these existing approaches.
2.The literature review in the introduction is overly brief and cites only one reference. Moreover, it contains inaccurate statements—for example, “MRI partitions the brain into regions of interest (ROIs)” is incorrect and should be more precisely described.
3. The references are outdated and lack recent related work. The manuscript does not include any papers from 2025 and cites only one paper from 2024. Furthermore, the experimental comparisons fail to include recent state-of-the-art methods.
4. The authors claim to achieve state-of-the-art (SOTA) performance; however, on the ABIDE dataset, only two metrics are the best, and on the ADNI dataset, only one metric is the best. The performance claims should therefore be stated more cautiously. In addition, the experimental result analysis is rather simple and lacks depth.
5. Some mathematical formulations are not clearly explained. For example, in Equation (1), the meaning and dimensionality of $e_i $ are not defined, which negatively impacts readability and reproducibility.

**Questions:**

1. The dataset split in the experiments (training/validation/test = 2:1:7) is unusual. Please explain the rationale and justification for this non-standard partitioning, and discuss how it affects result stability and comparability (e.g., whether it is consistent with prior work and whether it influences performance outcomes).
2. In the computation of structural similarity, correlation values are divided into NC/UC/PC categories with a threshold of 𝜃=0.3 , and coefficients $𝛼_1 $ =2, $𝛼_2 $ =0.5, $𝛼_3 $ =−1, $𝛼_4 $ =−0.5 are used for different category combinations. Please clarify how these hyperparameters (especially θand the αvalues) were determined—were they selected via systematic search, cross-validation, or heuristics? A sensitivity analysis or justification of these parameter choices would be valuable.
3. The experimental analysis is insufficient. It is recommended to expand this section by discussing reasons for the method’s performance on different metrics, and analyzing under which types of samples or network structures the method performs better or worse. Such analysis would help clarify the method’s applicability and limitations.
4. The conclusion or discussion section should explicitly address the limitations of the proposed method.

---

### Official Review · Reviewer_wxQz · 2025-10-24

**Soundness:** 1
**Presentation:** 2
**Contribution:** 2
**Rating:** 2
**Confidence:** 4

**Summary:**

This paper proposes SSL-BN, a self-supervised framework designed for brain network analysis, aiming to overcome the challenge of designing graph augmentations for fully connected brain networks. SSL-BN constructs a Dataset Graph using a structural similarity measure among graph instances and performs embedding dispersion and contrastive refinement without explicit data augmentation.

**Strengths:**

1. The paper addresses an important problem in self-supervised learning for brain networks, aiming to avoid augmentation schemes unsuitable for fully connected graphs.
2. The methodology is mathematically well-specified and integrates concepts from contrastive learning and structural graph similarity in an elegant way.

**Weaknesses:**

1. Overclaiming novelty of SSL for brain networks. The paper claims SSL-BN is the first self-supervised learning framework for brain networks. This is inaccurate. Numerous SSL methods for brain networks already exist and have been published before 2025 [1-4]. These works should be cited and discussed to properly position SSL-BN in the literature. The lack of comparison to these methods makes the novelty claim unconvincing. Moreover, the baselines in Table 1 only include generic SSL methods and a single brain network model. The comparison should be expanded to include contrastive-based brain network methods [5–7] for fairness.
2. Citation formatting issues. The citation style throughout the paper is inconsistent and sometimes confusing — e.g., missing differentiation between parenthetical and textual citations (“He et al. (2020)” vs. “(He et al., 2020)”). The authors should carefully revise according to ICLR guidelines to maintain clarity and professionalism.
3. Equation–figure inconsistency. The notations in Figure 1 do not align with those in the main text. For instance, the symbol \mathbb{R}  appearing in Figure 1 does not correspond to any definition in the methodology section.
4. Train/validation/test split issue. The dataset split ratio of 2:1:7 seems highly imbalanced. Given that ABIDE contains only ~1,000 subjects, the training set would include merely ~200 samples, which is prone to severe overfitting — consistent with the high training accuracy observed in related work. A more standard split (e.g., 8:1:1) would yield more stable evaluation and allow better comparison with prior literature.
5. Weak performance on ADNI dataset. The performance of SSL-BN on ADNIis worse than simple models such as GCN and GAT, which questions the generalization capability of the proposed pretraining strategy.
6. Lack of interpretability analysis. The paper focuses solely on numerical performance without offering interpretability analyses (e.g., salient ROI identification, attention heatmaps, or network-level visualization). As SSL-BN claims to learn meaningful structural representations, such analyses are essential to demonstrate biological plausibility and robustness.
7. Lack of complexity analysis.

[1] A Generative Self-Supervised Framework using Functional Connectivity in fMRI Data. NeurIPS 2023 Temporal Graph Learning Workshop
[2] PTGB: Pre-Train Graph Neural Networks for Brain Network Analysis. CHIL 2023
[3] BrainUSL: Unsupervised Graph Structure Learning for Functional Brain Network Analysis. MICCAI 2023
[4] BGCSL: An unsupervised framework reveals the underlying structure of large-scale whole-brain functional connectivity networks. CMPB 2024
[5] A-GCL: Adversarial graph contrastive learning for fMRI analysis to diagnose neurodevelopmental disorders. MIA 2023
[6] Contrastive Graph Pooling for Explainable Classification of Brain Networks. TMI 2024
[7] Contrasformer: A Brain Network Contrastive Transformer for Neurodegenerative Condition Identification. CIKM 2024

**Questions:**

See weaknesses.

---

### Official Review · Reviewer_rvpL · 2025-10-31

**Soundness:** 2
**Presentation:** 3
**Contribution:** 2
**Rating:** 4
**Confidence:** 4

**Summary:**

The paper proposes SSL-BN, a self-supervised framework for brain-network (fMRI) diagnosis that avoids graph augmentations. It first computes a structural similarity on a dataset graph to relate samples. Then it performs a dispersion step that L2-normalizes embeddings and scatters them on a unit hypersphere. Finally, it applies a non-trainable linear contrastive refinement whose parameters combine structural similarity with pairwise angular relations. BrainNetTF is used as encoder. Experiments on ABIDE (with additional ADNI results in the appendix) compare against supervised GNNs, graph SSL, and SimCLR/MoCo.

**Strengths:**

1. Tailored to Brain Networks. The framework addresses unique challenges of dense, weighted brain graphs by leveraging structural similarities.

2. Augmentation-Free and Simple Design. Avoiding complex augmentations and using a single loss function simplifies pre-training, improving efficiency for label-scarce neuroimaging tasks.

**Weaknesses:**

1. Insufficient Generalization and Out-of-Distribution Evaluation. ABIDE is multi-site data, but only random 2:1:7 splits with 5 repetitions are performed, without Leave-One-Site-Out (LOSO) or cross-dataset transfer (e.g., ABIDE - ADNI) to support the claim that “structural similarity brings stronger domain/site robustness.”

2. Insufficient Literature Review. The related work section primarily cites studies from several years ago (e.g., up to 2022–2023), with limited coverage of more recent advancements in the past two years (2024–2025). For instance, it overlooks emerging self-supervised learning methods tailored for fMRI brain networks, such as diffusion-augmented graph contrastive learning (GCDA) or personalized functional network generation via self-supervised deep learning.

3. Lack of Comparisons with Stronger SSL Baselines. The baselines like SimCLR and MoCo rely on random masking augmentations (which the authors acknowledge struggle to produce “similar samples” for brain networks), but the paper misses contrasts with more advanced, brain-network-specific SSL methods (e.g., self-supervised frameworks for brain connectivity structure learning or fine-tuned BrainNetTF with SSL pre-training on large fMRI cohorts). This limits the evaluation’s rigor.

4. Comparable or Underwhelming Experimental Results. Performance is often comparable to or even worse than baselines in some cases, particularly in the supplementary ADNI experiments, where only one metric exceeds the baseline, raising questions about the method’s robustness across datasets.

5. Inadequate Discussion of Complexity and Scalability. Constructing the similarity matrix S requires computing over n\times n sample pairs (O(n^2) overall, even with matrix operations enabling parallelism), and training approximates the dataset graph via mini-batches. However, the paper lacks analyses of time/memory usage on larger-scale data.

6. Missing Hyperparameter Analysis. The threshold θ=0.3 is described as the “best from multiple experiments,” but no sensitivity analysis or justification for its selection is provided, leaving readers unclear on the model’s robustness to parameter variations.

7. Lack of Interpretability. No visualizations of learned embeddings, biomarkers, or analyses linking structural similarities to neurobiological insights, reducing clinical value.

**Questions:**

1. How was the structural similarity measure validated biologically (e.g., does it align with known brain connectivity patterns)? Ablations on alternatives would help.

2. Given the inspiration from SGRL, what specific adaptations make SSL-BN uniquely suited for brain networks beyond the dataset graph?

3. How sensitive are results to the structural-similarity threshold(s) and the weights in the refinement matrix? Please include curves/heatmaps.

---

### Official Review · Reviewer_NX2c · 2025-11-01

**Soundness:** 2
**Presentation:** 1
**Contribution:** 2
**Rating:** 2
**Confidence:** 4

**Summary:**

SSL-BN proposes an augmentation-free SSL framework for fMRI brain networks. It builds a dataset-level structural similarity $\tilde S$ by discretizing ROI–ROI correlations (NC/UC/PC), first disperses normalized embeddings on the unit sphere via a dispersion loss, then applies a non-trainable linear refinement $E'=(\tilde S \odot \Omega)E$ with angle-aware gating $\Omega$. Using BrainNetTF on correlation matrices, the method pretrains without augmentations and yields modest AUROC gains on ABIDE; ablations indicate that both dispersion and refinement are necessary.

**Strengths:**

- **Domain-specific structural similarity on fMRI correlations.** The paper leverages neuroimaging prior knowledge by discretizing correlation signs (NC/UC/PC) to build a dataset graph and drive representation refinement without brittle augmentations. This is a clear, problem-matched design for dense, weighted brain connectivity.

- **Simple, augmentation-free SSL pipeline with interpretable components.** Training uses a single dispersion loss; refinement is a fixed linear operator $(\tilde S \odot \Omega)$, making the approach easy to implement and potentially more stable than augmentation-heavy objectives.

- **Empirical gains with informative ablations.** On ABIDE, SSL-BN improves over BrainNetTF and several GNN/graph-SSL/contrastive baselines; ablations show performance drops when removing dispersion or refinement or when using only $\tilde S$ or only $\Omega$.

**Weaknesses:**

1) **Insufficient coverage of recent SSL and baselines**
The paper claims that there are no existing self-supervised methods designed for brain networks, but this assertion is not substantiated. Recent SSL, including graph-based and contrastive methods targeting brain networks, is missing from both the discussion and the baselines, and it is needed to highlight and discuss the differences from recent methods. Moreover, describing BrainNetTF as state-of-the-art seems outdated, as more recent and advanced models for brain-network analysis have been introduced. A more comprehensive literature search and inclusion of recent brain-graph–targeted SSL approaches as baselines are needed to properly contextualize the contribution.

```
[1] Luo, Xuexiong, et al., "An interpretable brain graph contrastive learning framework for brain disorder analysis.", WSDM 2024.
[2] Peng, Liang, et al., "GATE: Graph CCA for temporal self-supervised learning for label-efficient fMRI analysis.", IEEE TMI 2022.
[3] Yang, Yi, Hejie Cui, and Carl Yang., "PTGB: Pre-Train Graph Neural Networks for Brain Network Analysis.", CHIL 2023.
[4] Behrouz, Ali, and Farnoosh Hashemi., "Brain-mamba: Encoding brain activity via selective state space models.", CHIL 2024.
[5] Li, Ying, et al., "Self-Supervised Learning to Unveil Brain Dysfunctional Signatures in Brain Disorders: Methods and Applications." Health Data Science 2025.
```

2) **Lack of modeling description and notation**
The paper explains that $f_i$ is transformed into $g_i$ and then encoded as $e_i$, but the description of this process is unclear. It is not specified how the 4D fMRI data $f_i \in \mathbb{R}^{T \times X \times Y \times Z}$ are converted at each stage or what the exact data dimensions are after each transformation.

3) **Scalability and unlabeled-data utilization under SSL.**
The paper states that supervised GNN baselines use a 2:1:7 split, but it is unclear whether the same proportion was used for SSL pretraining. If so, it raises the question of why a larger portion of unlabeled data was not utilized for pretraining. Since the main strength of SSL lies in exploiting large unlabeled datasets, limiting training to a small subset (e.g., only part of ABIDE) is difficult to justify. In addition, the SSL baselines employ contrastive learning methods that generally benefit from large batch sizes; it would be important to know whether SSL-BN maintains its advantage under different batch-size settings. If such ablation experiments exist, presenting them would strengthen the claim.

4) **Sensitivity of the structural similarity threshold $\theta$**
The method fixes $\theta=0.3$ based on experimentation, but no explicit sensitivity analysis is provided. The paper mentions that $\theta=0.3$ yields the best performance, yet it does not show the results of the extensive experiments that led to this choice. If ablation studies over different $\theta$ values were conducted, including them would clarify how sensitive the method is to this parameter. Also, it would also be helpful to report whether the same $\theta$ value is optimal for both ABIDE and ADNI datasets.

**Minor**
1. The main text states that experiments were conducted on both ABIDE and ADNI, but only ABIDE results are presented. A brief summary of the ADNI results from Appendix A.1 should be included in the main text for completeness.
2.  Appendix typo: “ABIDE DATSAET” → “ABIDE DATASET” (Section A.2, A.3 and table caption).
3. Table5 typo: “coor means correlation matrix” → “corr means correlation matrix”.

**Questions:**

**Q1)** What is the rationale for using the 2:1:7 data split? Was this configuration specifically considered when designing the SSL pretraining process?

**Q2)** Are there ablation results for different values of $\theta$?

**Q3)** Can the authors provide qualitative evidence of the learned representations, such as UMAP or t-SNE visualizations of contrastively aligned embeddings, to demonstrate whether the model captures meaningful structure for downstream tasks?

**Q4)** External validation: Are pretraining and fine-tuning performed within the same dataset (ABIDE→ABIDE; ADNI→ADNI)? If so, have the authors explored a cross-dataset transfer setting (e.g., ABIDE pretrain → ADNI fine-tune)? Such an experiment could not only help compensate for limited data when training the Transformer encoder but also better demonstrate generalization under label scarcity and dataset shift, which is a central motivation for SSL.

---

### Note · Authors · 2025-11-17

**Comment:**

We thank all the reviewers for their thorough evaluations. We have learned a great deal from the identified limitations of our work and have ultimately decided to withdraw the submission. We plan to revise and improve the work in the future and, if possible, pursue publication at a later stage.

**Withdrawal Confirmation:**

I have read and agree with the venue's withdrawal policy on behalf of myself and my co-authors.